# Suppression of autophagic activity by Rubicon is a signature of aging

Shuhei Nakamura [1,2,3], Masaki Oba[4,5], Mari Suzuki [4], Atsushi Takahashi[6], Tadashi Yamamuro[1], Mari Fujiwara[1], Kensuke Ikenaka[7], Satoshi Minami[6], Namine Tabata[8,9,10], Kenichi Yamamoto[11], Sayaka Kubo[1,2], Ayaka Tokumura[1], Kanako Akamatsu[1], Yumi Miyazaki[1,2], Tsuyoshi Kawabata[1,12], Maho Hamasaki[1,2], Koji Fukui[5], Kazunori Sango[4], Yoshihisa Watanabe[13], Yoshitsugu Takabatake [3], Tomoya S. Kitajima [8,9,10], Yukinori Okada [11], Hideki Mochizuki [7], Yoshitaka Isaka[6], Adam Antebi[14,15] & Tamotsu Yoshimori[1,2]

Autophagy, an evolutionarily conserved cytoplasmic degradation system, has been implicated as a convergent mechanism in various longevity pathways. Autophagic activity decreases with age in several organisms, but the underlying mechanism is unclear. Here, we show that the expression of Rubicon, a negative regulator of autophagy, increases in aged worm, fly and mouse tissues at transcript and/or protein levels, suggesting that an age-dependent increase in Rubicon impairs autophagy over time, and thereby curtails animal healthspan. Consistent with this idea, knockdown of Rubicon extends worm and fly lifespan and ameliorates several age-associated phenotypes. Tissue-specific experiments reveal that Rubicon knockdown in neurons has the greatest effect on lifespan. Rubicon knockout mice exhibits reductions in interstitial fibrosis in kidney and reduced α-synuclein accumulation in the brain. Rubicon is suppressed in several long-lived worms and calorie restricted mice. Taken together, our results suggest that suppression of autophagic activity by Rubicon is one of signatures of aging.

---

[1] Department of Genetics, Graduate School of Medicine, Osaka University, Osaka 565-0871, Japan. [2] Department of Intracellular Membrane Dynamics, Graduate School of Frontier Biosciences, Osaka University, Osaka 565-0871, Japan. [3] Institute for Advanced Co-Creation Studies, Osaka University, Osaka 565-0871, Japan. [4] Diabetic Neuropathy Project, Department of Sensory and Motor Systems, Tokyo Metropolitan Institute of Medical Science, Setagaya, Tokyo 156-8506, Japan. [5] Department of Bioscience and Engineering, Shibaura Institute of Technology, Saitama 337-8570, Japan. [6] Department of Nephrology, Graduate School of Medicine, Osaka University, Osaka 565-0871, Japan. [7] Department of Neurology, Graduate School of Medicine, Osaka University, Osaka 565-0871, Japan. [8] Laboratory for Chromosome Segregation, RIKEN Center for Biosystems Dynamics Research (BDR), Kobe 650-0047, Japan. [9] Laboratory of Molecular Cell Biology and Development, Graduate School of Biostudies, Kyoto University, Kyoto 606-8501, Japan. [10] Laboratory of Biomolecular Informatics, Graduate School of Science, Osaka University, Osaka 560-0043, Japan. [11] Department of Statistical Genetics, Graduate School of Medicine, Osaka University, Osaka 565-0871, Japan. [12] Department of Stem Cell Biology, Atomic Bomb Disease Institute, Nagasaki University, Nagasaki 852-8523, Japan. [13] Department of Basic Geriatrics, Graduate School of Medical Science, Kyoto Prefectural University of Medicine, Kyoto 602-8566, Japan. [14] Department of Molecular Genetics of Ageing, Max Planck Institute for Biology of Ageing, Cologne 50931, Germany. [15] Cologne Excellence Cluster on Cellular Stress Responses in Aging Associated Diseases (CECAD), University of Cologne, Cologne 50931, Germany. These authors contributed equally: Shuhei Nakamura, Masaki Oba. Correspondence and requests for materials should be addressed to T.Y. (email: tamyoshi@fbs.osaka-u.ac.jp)

Macroautophagy (hereafter, autophagy) is an evolutionarily conserved intracellular membrane trafficking process in which double-membrane structures called autophagosomes sequester cytoplasmic materials and fuse with lysosomes, where their contents are degraded. Initially, autophagy was described as a bulk degradation system, but it has become clear that autophagy also selectively targets aggregated proteins, lipids, damaged organelles, and invading bacteria. By driving the degradation of a wide range of targets, autophagy maintains cellular homeostasis; consequently, dysfunction in autophagy has been implicated in many human diseases, including cancer, neurodegeneration, and metabolic disorders.

Recent evidence has shown that autophagy is also involved in animal aging. Autophagic activity decreases with age in many species[1–4]. Studies using several model organisms including *C. elegans* led to the discovery of several conserved longevity pathways, including mild reduction of insulin/IGF-1 signalling, calorie restriction, germline removal, reduced mitochondrial respiration and reduced TOR signalling. Importantly, all of these interventions activate autophagy and extend animal lifespan in a manner that depends on active autophagy, suggesting that autophagy is one of convergent mechanisms of many longevity pathways[5–9]. Moreover, overexpression of ATG5 in mice or neuronal Atg8 or Atg1 in *Drosophila* extends lifespan[4,10,11], although it remains unclear why simple overexpression of genes involved in autophagosome formation would activate autophagy. More recently, a knock-in gain-of-function point mutation in Beclin-1 which disrupt beclin1-BCL2 interaction and constitutively activates autophagy has been shown to extend lifespan in both female and male mice[12]. Although these accumulating evidences further manifest the positive correlation between activation of autophagy and longevity[13], yet our knowledge of the molecular mechanism by which autophagic activity declines with age is still limited[14].

Although many autophagy related genes are positive regulators of autophagy, we and others previously identified one of few negative regulators of autophagy, Rubicon (Run domain Beclin-1 interacting and cysteine-rich containing protein), as a Beclin 1 interacting protein. Especially, Rubicon inhibits autophagosome-lysosome fusion process as well as endocytic trafficking through binding to PI3K (class III phosphatidylinositol-3 kinase) complex[15,16]. Recently, we have found that Rubicon level is increased in association of autophagy impairment in livers of mice fed a high-fat diet, recapitulating NAFLD (nonalcoholic fatty liver disease)[17]. Hepatocyte specific Rubicon knockout mice displays significant improvement of liver steatosis and autophagy, indicating upregulation of Rubicon plays a pathogenic role in NAFLD[17]. Since the prevalence of NAFLD increases with age[18,19], we decided to examine the relationship between Rubicon and aging. In the current study, we found that Rubicon is increased in worm, fly and mouse tissues. Reduction of Rubicon extends lifespan in worm and female fly and ameliorates age-associated phenotype in worm, fly, and mouse tissues. These results suggest that increase of Rubicon could be evolutionarily conserved one of causes for age-dependent autophagy impairment.

## Results

### *C. elegans* Rubicon homologue negatively regulates autophagy.
To examine the role of Rubicon in aging, we searched for *C. elegans* Rubicon homologues and found that Y56A3A.16 has sequence similarity to human Rubicon (KIAA0226), human KIAA0226L (recently identified as Pacer)[20], and *Drosophila* Rubicon (*dRubicon*, CG12772)[21] (Supplementary Fig. 1a, b). In mammals, Rubicon inhibits autophagy, whereas Pacer is required for normal progression of autophagy. Therefore, we next sought to

determine which of these two proteins is more functionally similar to Y56A3A.16. Knockdown of Y56A3A.16 by RNAi significantly increased the number of GFP::LGG-1 dots, an autophagosome marker[5,22], in intestinal cells (Fig. 1a, b) as previously observed in Rubicon knockdown mammalian cells[15]. This increase was completely abolished by concomitant knockdown of *bec-1/Beclin1* or *atg-18/Atg18*, both of which function in autophagosome formation, suggesting that knockdown of Y56A3A.16 increased autophagic vacuoles (Fig. 1a, b and Supplementary Fig. 1c). To further discriminate whether knockdown of Y56A3A.16 increase or block autophagic activity, we used the recently developed transgenic worm expressing tandem fluorescent LGG-1 (mCherry::GFP::LGG-1)[1]. With this reporter, autophagosomes (AP) are visualised as punctae positive for both GFP and mCherry, while autolysosomes (AL) show only the mCherry due to quenching of GFP in the acidic environment of lysosomes. We observed knockdown of Y56A3A.16 significantly increases the numbers of autophagosomes and autolysosomes in intestines and pharyngeal muscles (Supplementary Fig. 2a–d), indicating that the pool size of autophagosome and autolysosomes are increased by the knockdown. Furthermore, to examine the autophagic flux, we injected either DMSO or lysosomal inhibitor, BafA in control or Y56A3A.16 knockdown worms and compared the number of GFP::LGG-1 puncta. Unexpectedly, we could not observe the significant difference between control and Y56A3A.16 knockdown worms in DMSO injected worms. The exact reasons are unknown, but the microinjection might cause the unknown stresses and dampen the effect of Y56A3A.16 knockdown. Nevertheless, we could clearly observed the statistically much increased numbers of GFP::LGG-1 puncta in Y56A3A.16 knockdown worms after BafA injection compared to control knockdown, suggesting that Y56A3A.16 knockdown increases the autophagic flux (Fig.1c and Supplementary Fig. 2e). Taken together with the fact that there is only one Rubicon homologue in zebrafish and *Drosophila* (Supplementary Fig. 1a)[21], our results imply that Y56A3A.16 represents the ancestral homologue of Rubicon; accordingly, hereafter we refer to Y56A3A.16 as RUB-1.

### Knockdown of *rub-1* extends lifespan.
Interestingly, gene expression analysis using RUB-1::EGFP transgenic worms in which EGFP is expressed in several tissues including neurons, intestine and pharynx and we observed RUB-1::EGFP was increased in older worms (Fig.1d). qRT-PCR analysis revealed that *rub-1* was increased with age at transcript levels (Fig. 1e). On the other hands, *rub-1* was downregulated in several long-lived mutants compared to wild type (Fig.1f). These results led us to hypothesise that the age-dependent increase of *Rubicon*, a negative regulator of autophagy, could be one of causal factors for age-dependent impairment of autophagy. Consistent with this idea, knockdown of *rub-1* by RNAi significantly extended wild-type worm lifespan (Fig. 1g and Supplementary Fig. 3a, b). The *rub-1* transcript level was decreased to ~40–50% of normal levels in these animals (Supplementary Fig. 3c–e). Knockdown of *rub-1* does not alter the pharyngeal pumping rates indicating that the bacteria containing RNAi is properly took in (Supplementary Fig. 3f). In mammalian cells, Rubicon has been shown to repress both autophagy and endocytic pathway including recycling of receptors such as transferrin receptor[15]. We saw that *rub-1* knockdown did not largely alter the localisation of hTFR (human transferrin receptor)::GFP in intestinal cells, suggesting that *rub-1* might not have obvious function in endocytosis in worms (Supplementary Fig. 2f). Importantly, the increase in longevity was completely abolished when the autophagy regulators *bec-1/Beclin1*, *unc-51/ULK1* and *atg-18/ATG18* were knocked down by RNAi along with *rub-1* (Fig. 1g and Supplementary Fig. 3a, b),

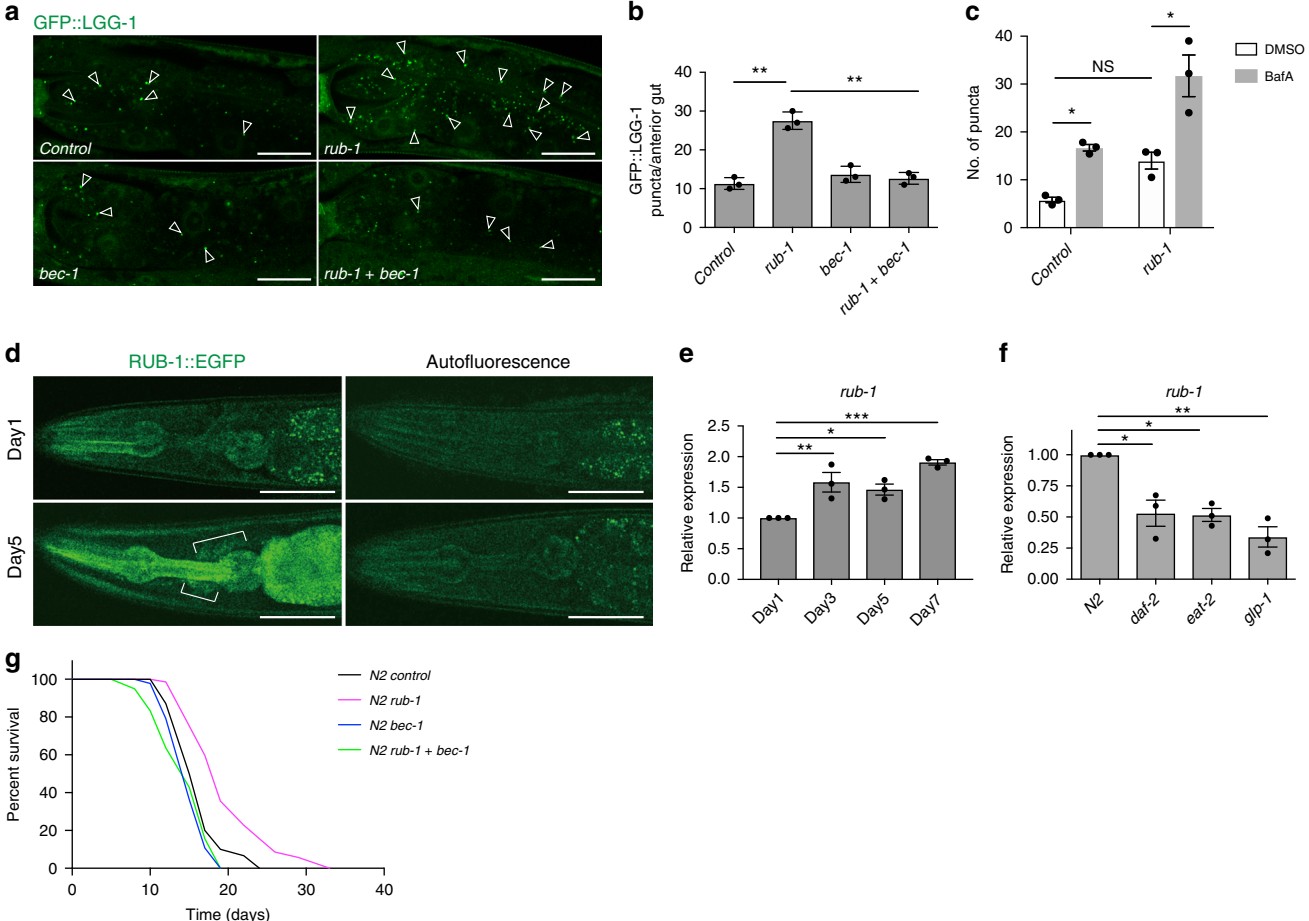

**Fig. 1** *C. elegans* Rubicon regulates lifespan via modulating autophagy. **a** Autophagosomes labelled with GFP::LGG-1 (open arrowheads) were more abundant in *rub-1* knockdown, but concomitant knockdown of *bec-1/Beclin 1* abolished the increment. Anterior intestines of GFP::LGG-1 transgenic worms at L4 stage were shown. Each RNAi is indicated. Knockdown was conducted from egg onward. **b** Quantification of GFP::LGG-1 puncta in anterior intestines under each knockdown condition. Values represent means ± s.e.m. (n = 3). P value (**P < 0.01) was determined by one-way ANOVA with Tukey's test. **c** No. of GFP::LGG-1 is more increased by *rub-1* knockdown 2 h after BafA injection compared to control knockdown, indicating the autophagic flux is increased by *rub-1* knockdown. Values represent means ± s.e.m. (n = 3). P value (*P < 0.05, **P < 0.01) was determined by one-way ANOVA with Tukey's test. The representative picture of DMSO or BafA injected GFP::LGG-1 worms were shown in Supplementary Fig. 2e. **d** Representative pictures of transgenic worm expressing RUB-1::EGFP showing that the fluorescence in anterior region including neuronal cells (brackets) were increased at day 5 compared to day 1. All images were taken together with N2 showing autofluorescence at same exposure times. **e** qRT-PCR analysis showing *rub-1* expression at day 1, 3, 5 and 7 in wild-type worms, indicating that *rub-1* is upregulated from day3 onward. Mean ± s.e.m. from three independent experiments are depicted and are normalised to wild-type N2 day1 samples. P value (*P < 0.05, **P < 0.01, ***P < 0.001) was determined by one-way ANOVA with Tukey's test. **f** qRT-PCR analysis showing *rub-1* expression in several long-lived strains including *daf-2(e1370)*, *eat-2(ad465)*, and *glp-1(e2141)* at day1 adult stage. Mean ± s.e.m. from three independent experiments are depicted and are normalised to wild-type N2 day1 samples. Germline-less phenotype of *glp-1* was induced by exposure to elevated temperature (25 °C) for 2 days. P value (*P < 0.05, **P < 0.01) was determined by one-way ANOVA with Tukey's test. **g** Knockdown of *rub-1* extends wild-type lifespan. Longevity conferred by *rub-1* was abolished by concomitant knockdown of *bec-1/Beclin1*. Knockdown was conducted from egg onward. Log-rank test was conducted for statistical analysis. See Supplementary Data 1 for details and repeats. Scale bars, 20 μm (**a**); 50 μm (**d**)

indicating that the longevity enhancement conferred by knockdown of *rub-1* is dependent on autophagic activity. Conversely, we found that the overexpression of *rub-1* shortened wild-type worm lifespan (Supplementary Fig. 4a, b). Together our results suggest *rub-1* regulates the lifespan by modulating autophagic activity.

In worms, several different longevity pathways and corresponding long-lived mutants have been well characterised; these pathway include reduction of insulin/IGF-1 signalling (*daf-2*), germline removal (*glp-1*), calorie restriction (*eat-2*) mitochondrial dysfunction (*isp-1*) and downregulation of TOR signalling. These animals show increased autophagy and the longevity depends on active autophagy[8,23]. Therefore we checked the genetic interaction between *rub-1* and some of these longevity pathways. *rub-1*

knockdown did not further extend lifespan of *daf-2*, *glp-1*, *isp-1* and *eat-2* animals (Supplementary Fig.4c–f). Although technically we did not check all of these longevity mutants, *rub-1* knockdown did not change the numbers of autophagosomes in tested long-lived animals (Supplementary Fig.5). Interestingly, *rub-1* knockdown additionally extended lifespan of TOR knockdown animals (Supplementary Fig.4g), implying these two genes act at least partly in parallel.

**Knockdown of *rub-1* ameliorates age-associated phenotypes.** We next investigated whether knockdown of *rub-1* would impact other age-onset phenotypes. Worms expressing polyglutamine (polyQ)-YFP in body wall muscle exhibit age-onset aggregation of

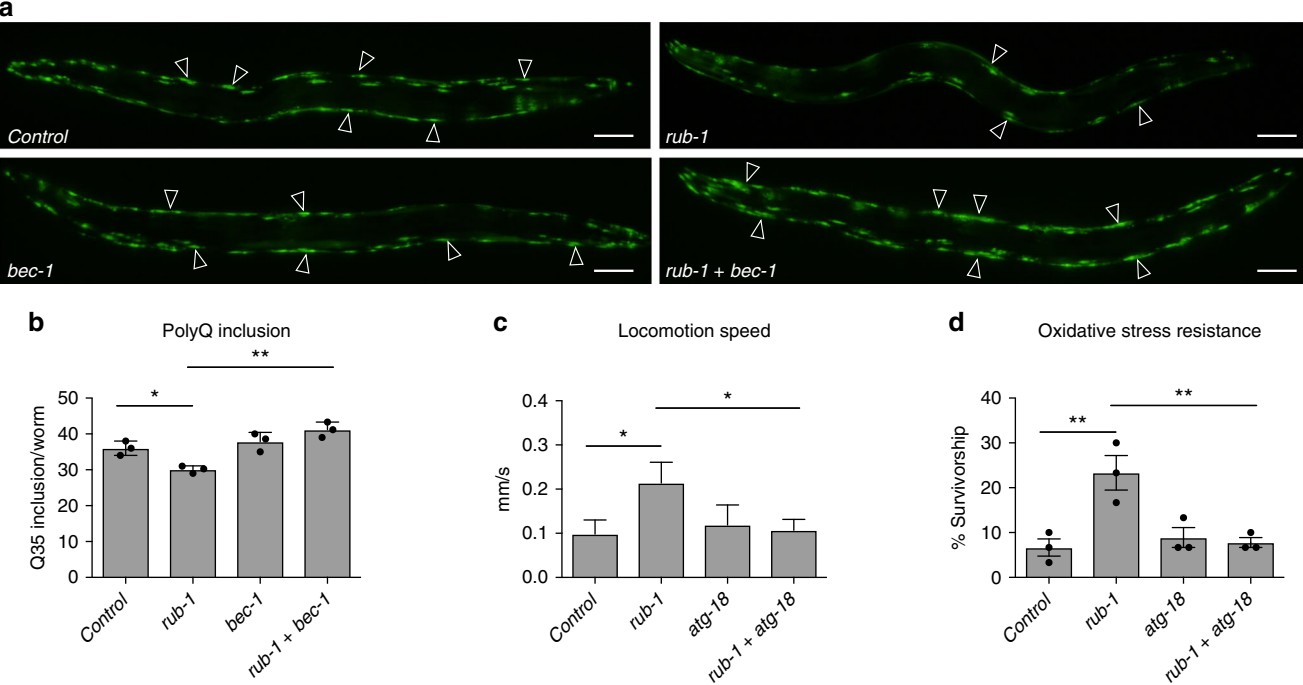

**Fig. 2** Rubicon knockdown ameliorates age-dependent phenotypes in worms. **a** Transgenic worms expressing polyQ35::YFP in body wall muscle exhibited age-dependent accumulation of polyQ inclusions (open arrowheads) at day5 stage. *rub-1* knockdown decreased the accumulation of inclusions, and concomitant knockdown of *bec-1* reverted the phenotype. Each RNAi is indicated. Knockdown was conducted from egg onward. **b** Number of polyQ inclusions per worm in **a**. More than 20 worms were analysed in each experiment, and the experiments were repeated three times. Values represent means ± s.e.m. (*n* = 3). *P* value (*$P < 0.05$, **$P < 0.01$) was determined by *t*-test. **c** Locomotion speed, calculated from multi-worm tracking analysis at day5. *rub-1* knockdown worms maintain high locomotion activity. Knockdown was conducted from egg onward. Values represent means ± s.e.m. (control, *n* = 12; *rub-1*, *n* = 8; *atg-18*, *n* = 8; *rub-1* + *atg-18*, *n* = 13). *P* value (*$P < 0.05$) was determined by *t*-test. The representative picture of the tracking is shown in supplementary fig. 6a. **d** *rub-1* knockdown increased the oxidative stress resistance in an autophagy dependent manner. Survivorship of wild-type day 1 worms subjected to indicated RNAi after 4.4 mM $H_2O_2$ treatment for 2 h. Knockdown was conducted from egg onward. Values represent means ± s.e.m. (*n* = 3). *P* value (**$P < 0.01$) was determined by *t*-test. Scale bars 50 μm (**a**)

the polyQ fusion protein[24]. Knockdown of *rub-1* reduced the level of aggregation, whereas concomitant knockdown of *bec-1*/ *Beclin 1* reverted the reduction, indicating that knockdown of *rub-1* decreases aggregation of polyQ fusion protein in an autophagy-dependent manner (Fig. 2a, b). Locomotion activity goes down with age, and velocity correlates well with longevity[25]. A multi-worm tracking system allowed us to monitor the tracks of individual worms and calculate their locomotion speed[26]. This analysis revealed that *rub-1* knockdown slowed down the age-dependent decline in locomotion activity in an autophagy-dependent manner (Fig. 2c and Supplementary Fig. 6a). Many long-lived animals often show the resistances against several stresses including oxidative stresses and heat stresses. We found that knockdown of *rub-1* increases the oxidative stress resistance in an autophagy-dependent manner (Fig. 2d). On the other hands heat stress resistance is not changed (Supplementary Fig. 6b), indicating that *rub-1* knockdown confers the different downstream output with regard to the stress resistance.

**Tissue specific roles of *rub-1* regulating lifespan**. Next we also investigated which tissues of *rub-1* are responsible for lifespan regulation in *C. elegans*. For this purpose, we used tissue-specific RNAi-sensitive strains, including TU3401 (neuron-specific, using the *sid-1* system)[27], NR350 (muscle), VP303 (intestine) and NR222 (hypodermis), for *rub-1* knockdown (Fig. 3a and Supplementary Fig. 7a–c)[28]. Surprisingly, knockdown of *rub-1* in neuronal cells extended lifespan most efficiently (Fig. 3a). Hypodermal and intestinal knockdown of *rub-1* extended

lifespan to a lesser extent but significantly (Supplementary Fig. 7a-c). Because neuronal cells are normally resistant to RNAi knockdown, the lifespan extension conferred by *rub-1* knockdown in wild-type worms might reflect combinatorial effects from hypodermis, intestine, and other unidentified tissues (Fig. 1g and Supplementary Fig. 3a, b). We observed that *atg-18*/*WIPI* was required for longevity conferred by *rub-1* knockdown in neuronal cells (Fig. 3b and Supplementary Fig. 7d). To check the autophagic flux in neuronal cells, we injected DMSO or BafA in control or *rub-1* neuron specific knockdown worms which express neuron specific GFP::LGG-1. Unexpectedly, we could not detect the significant increase of GFP::LGG-1 dots after BafA injection in control knockdown worms. It could be possible that the basal autophagy is relatively low in neuronal cells and is difficult to detect the flux at this condition. Nevertheless, we observed the significant increase of GFP::LGG-1 dots after BafA injection in *rub-1* knockdown worms, indicating the autophagic flux is increased in these worms. (Fig. 3c, d). These results together imply that the lifespan extension by *rub-1* knockdown could be due to activation of autophagy.

**Knockdown of *dRubicon* extends lifespan in fly**. To know if the findings in *C. elegans* is conserved in other species, we first checked expression of *Drosophila* Rubicon (dRubicon) and observed the dRubicon was significantly upregulated in middle-aged female fly (Day 14) compared to young fly (Day 2) (Fig. 4a, b and Supplementary Fig. 8a, b). To assess autophagic activity, we applied tandem fluorescent tagged Atg8a which are widely used

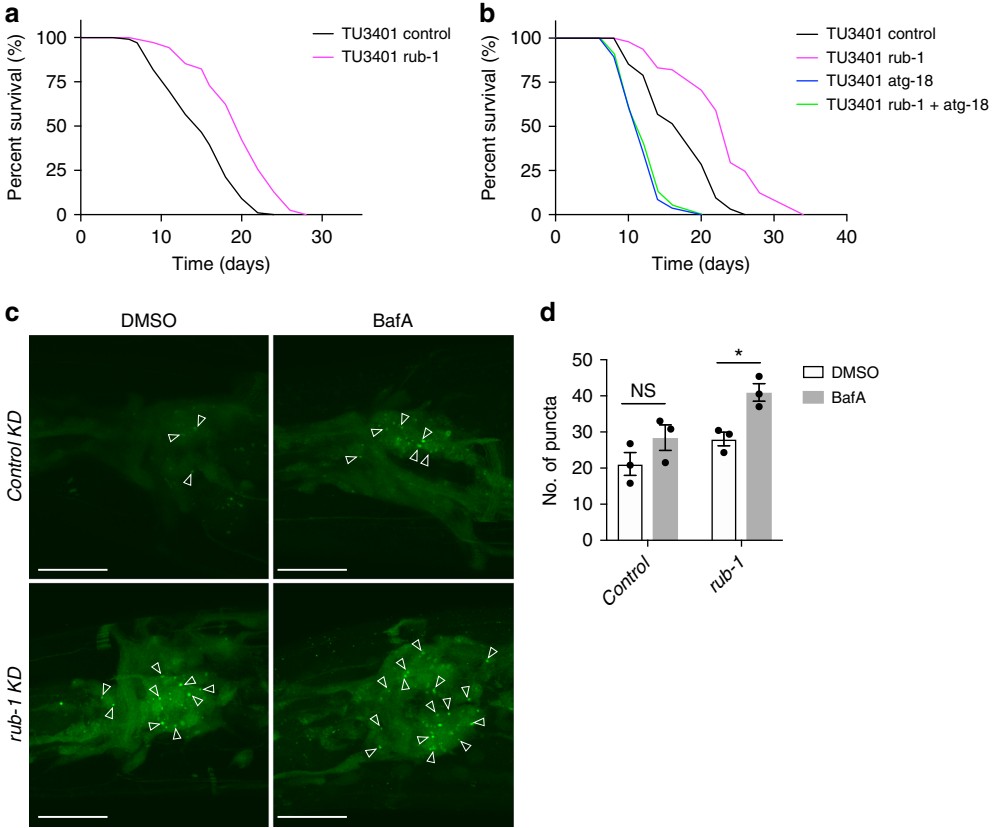

**Fig. 3** Tissue specific roles of Rubicon contributing to lifespan regulation in worms. **a, b** Neuronal knockdown of *rub-1* extended lifespan efficiently. The longevity by *rub-1* knockdown was abolished by concomitant *atg-18* knockdown. TU3401(neuron specific RNAi sensitive strain) was used and knockdown was conducted from egg onward. See Supplementary Data 1 for details and repeats. **c** Neuron specific knockdown of *rub-1* significantly increased no. of GFP::LGG-1 2 h after BafA injection compared to DMSO injection in neuron, while control knockdown did not, indicating autophagy flux was increased by *rub-1* knockdown in neuron. Newly generated strains by crossing between MAH215 and TU3401 were used and images were taken at day 1 adult stage. Knockdown was conducted from egg onward. Arrowheads indicate GFP::LGG-1 puncta. **d** Quantification of GFP::LGG-1 puncta in neuron from **c**. Values represent means ± s.e.m. (*n* = 3). *P* value (**P* < 0.05) was determined by *t*-test. Scale bars, 20 μm (**c**)

in fly to monitor autophagic activity[29]. Systemic knockdown of *dRubicon* significantly increased steady-state autophagosome and autolysosome pools in the brain (Fig. 4c, d). Although in most cases researchers in fly field concluded autophagy activation based on these similar phenotypes, we could not completely rule out the possibility of autophagy inhibition due to, for instance, impaired lysosomal degradation[30]. Importantly, whole body *dRubicon* knockdown slightly but significantly extended the lifespan in female specific manner (Supplementary Fig. 8c–e). In addition, knockdown of *dRubicon* also improved compound eye degeneration in MJDtr-Q78 (a truncated form of MJD/SCA3 containing an expanded glutamine tract) polyQ disease model female flies (Fig. 4e, f). Rapamycin, an inhibitor of mTOR kinase treatments increased the lifespan more in female mice than in male. Thus, our observation might reflect the sexual dimorphic contribution of autophagy to the animal lifespan.[31–33] Furthermore, similar to worms, neuron specific knockdown of *dRubicon* efficiently extended lifespan (both maximum and median lifespan) in female fly (Fig. 4g and Supplementary Fig. 8f). In addition, neuron specific *dRubicon* knockdown decreased the polyQ inclusions in female brain neurons (Fig. 4h, i) and mitigated the age-associated decline of locomotor function accelerated by the polyQ cytotoxicity (Fig. 4j and Supplementary Fig. 8g). These results indicate the conserved role of Rubicon, knockdown of which extends lifespan and reduces the aggregates of aggregation-prone proteins via modulating autophagic activity between worm and female fly.

**Rubicon deletion prevents age-associated phenotypes in mice.** Moreover, similar to worm and fly, we found that the Rubicon transcript and protein level in kidney and liver were higher in older animals (20 months of age) than in juveniles (2 months) in mice (Fig. 5a, b, and Supplementary Fig. 9a–c). Interestingly, mice subjected to 9-month calorie restriction exhibited reduced Rubicon protein levels both in liver and kidney (Supplementary Fig. 9 d, e), suggesting that Rubicon could be regulated downstream of lifespan extending condition similar to worms. To understand the role of Rubicon regarding animal aging in mice, we developed Rubicon systemic-knockout mice, which exhibit higher levels of LC3-II and reduced levels of the autophagic substrate p62, suggesting activation of basal autophagy (Fig. 5c). Progressive fibrosis is a histological hallmark of aging kidney[34]. Rubicon systemic-knockout mice exhibited a reduction in age-associated fibrosis, as determined by immunohistochemistry for collagen I (Fig. 5d, e). This phenotype was confirmed by qRT-PCR detection of mRNAs encoding fibrotic markers (Fig. 5f). Autophagy has been implicated in several age-associated neurodegenerative disorders, including Parkinson's disease. In sporadic and familial Parkinson's disease, affected neurons develop inclusions containing α-synuclein (α-Syn). Elevated amounts of this protein are sufficient to cause Parkinson's disease. It is suggested that overexpressed and misfolded aggregate-prone forms of α-Syn are degraded by autophagy[35–37]. Therefore, we investigated whether forced activation of autophagy by deletion of Rubicon in neurons would be sufficient to prevent aggregation of α-Syn in the brain. To explore

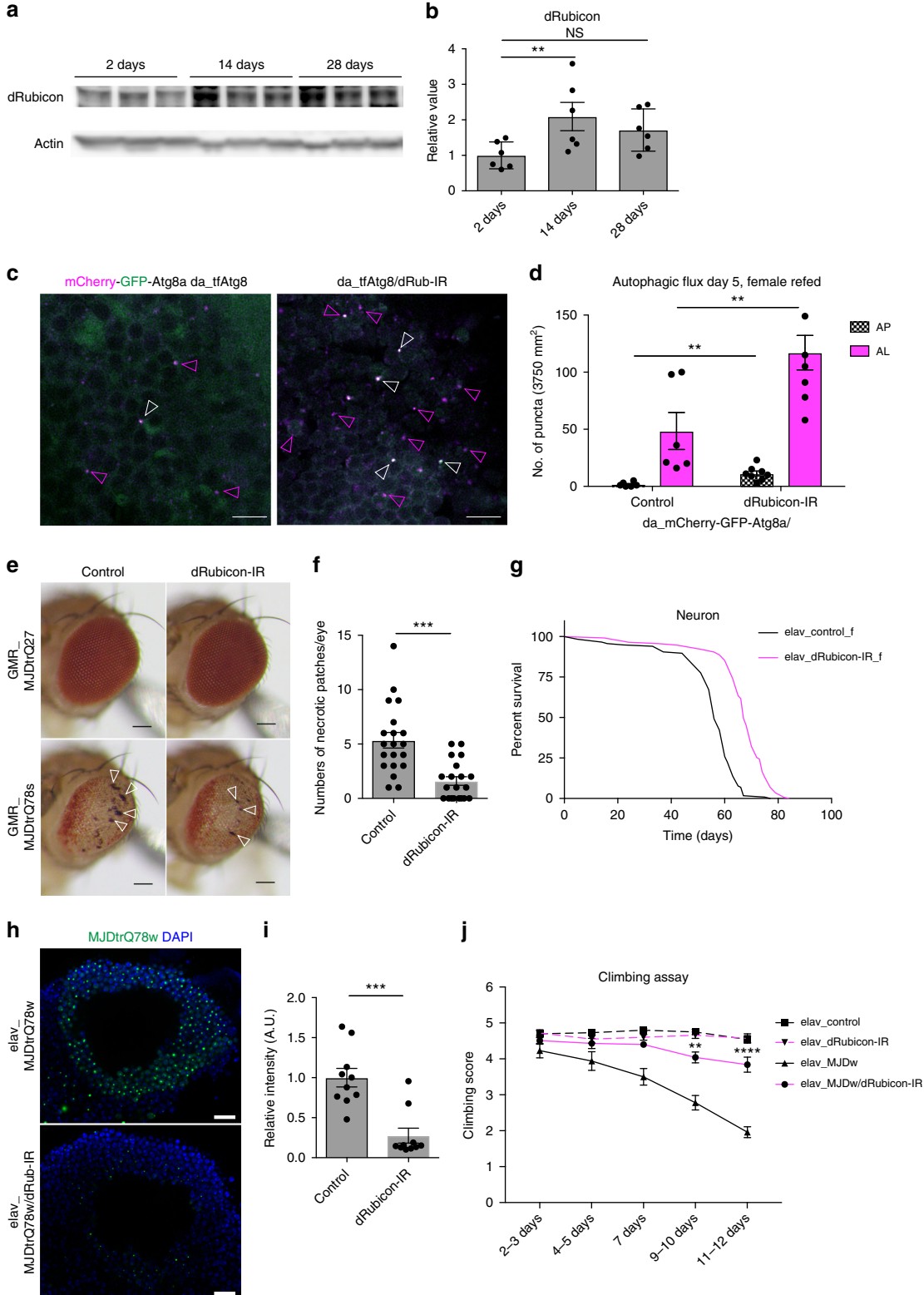

this possibility, we injected preformed α-Syn fibrils into the striatum of control and Rubicon neuronal (*Nestin-cre*) knockout mice as previously described[38,39], and compared the formation of Lewy body and Lewy neurite–like α-Syn inclusions after 10 months. Remarkably, the level of phosphorylated α-Syn positive signals was significantly reduced in neuron-specific Rubicon-knockout mice (Fig. 5g, h), suggesting that neuronal Rubicon deletion suppressed the expansion of α-synuclein pathology. Together with the results in worms and fly, these results suggest that neuronal knockdown of Rubicon is sufficient to ameliorate aging phenotypes in multiple organisms.

## Discussion

In this study, we report the novel role of Rubicon in lifespan regulation and aging. Rubicon was originally identified as a Beclin

**Fig. 4** *dRubicon* knockdown extends lifespan and ameliorates polyQ protein-induced toxicity in female fly. **a** dRubicon was increased in middle-aged female fly. **b** A quantification of dRubicon levels normalised by Actin in **a**. Values represent means ± s.e.m. ($n = 6$, each time points). $P$ value (*$P < 0.05$) was determined by One-way ANOVA with Tukey's test. **c** The autophagy assay in Kenyon cell layers of brains showing that dRubicon knockdown increased autophagosomes (AP, open white arrowheads) and autolysosomes (AL, open magenta arrowheads). **d** Quantification of mCherry-GFP-Atg8 puncta in **c**. The values represent the mean ± s.e.m. $P$ values (**$P < 0.01$) were determined by $t$-test ($n = 6$ for control, $n = 8$ for dRub-IR). **e** Compound eye degeneration was attenuated by *dRubicon* knockdown in flies expressing the expanded polyQ protein MJDtrQ78s (*GMR*-GAL4_MJDtrQ78s flies) compared to control. Open arrowheads indicate necrotic patches. **f** A quantification of the numbers of necrotic patches in *GMR*-GAL4_MJDtr-Q78s fly eyes in **e**. More than 20 eyes were analysed for each genotype. $P$ value (***$P < 0.001$) was determined by $t$-test. **g** Neuronal knockdown of *dRubicon* extends lifespan in female fly. Transgene expression of *dRubicon-IR* was induced by the pan-neuronal *elav*-GAL4 driver. See Supplementary Data 1 for details and repeats. **h** Neuronal knockdown of *dRubicon* decreased polyQ (MJDtrQ78w) inclusions in Kenyon cell layer of female fly brains. **i** Quantification of the polyQ protein intensity in **h**. 10 bilateral layers of Kenyon cell layers from 5 animals were analysed. $P$ value (***$P < 0.001$) was determined by $t$-test. **j** *dRubicon* knockdown in neurons ameliorated the locomotor dysfunction by polyQ expression in female fly. Locomotor function was evaluated by the climbing assay. Data represents mean ± s.e.m. $P$ values (*$P < 0.05$, **$P < 0.01$, ***$P < 0.0001$) were determined by two-way ANOVA with Tukey's test (*elav*_MJDtrQ78w vs. *elav*_MJDtrQ78w/Rubicon-IR). MJDtrQ78w expression level did not change between *elav*_MJDtrQ78w and *elav*_MJDtrQ78w/Rubicon-IR (Supplementary Fig. 8g). Scale bars, 5 μm (**c**); 100 μm (**e**); 10 μm (**h**)

1–binding protein that represses autophagy and endocytosis[15,16]. Here we characterised a role of Rubicon in different species and found that its expression increased in aged worm, female fly and mouse tissues for the first time. Our results suggest that the increase in Rubicon levels, which causes suppression of autophagic activity, curtails lifespan in worms and female fly and causally contributes to several age-onset phenotypes, including polyQ aggregation in worms and fly, fibrosis in mouse kidney and α-Syn accumulation in mouse brain.

Although we have shown that Rubicon knockdown activates autophagy, it remains unclear how Rubicon regulates autophagy. In our previous paper, we showed that the Beclin 1 complex containing Rubicon negatively regulates autophagy and endocytosis[15]. Specifically, Rubicon negatively regulates autophagosome–lysosome fusion steps. However, how this happens is still unclear and other unidentified mechanisms may contribute to the repression of autophagy.

What is the substrate of autophagy that contributes to longevity? Mitophagy, which contributes to mitochondrial homeostasis, is essential for the lifespan extensions conferred by several genetic and pharmacological treatments[40–43]. Thus, forced activation of autophagy by Rubicon knockdown could also activate mitophagy, and this possibility will be addressed in our future work. Lipophagy is also essential for longevity in *C. elegans*[6,44]. Indeed, we previously showed that liver-specific Rubicon knockout ameliorates liver steatosis in mice fed a high-fat diet[17]. This effect seems to be mediated by enhanced lipophagy resulting from Rubicon knockout. On the other hand, our results revealed that among cell types, Rubicon knockdown in neurons extended lifespan efficiently. Non-cell-autonomous neuronal regulation of animal lifespan has recently attracted a great deal of attention in the field[45,46]. In this context, enhanced autophagic activity might contribute to the secretion of systemic signals from neuronal cells;[47] the candidate systemic signals include miRNAs, exosomes, small molecules and hormones. Contrary, other group showed that neuronal inhibition of autophagy after the reproductive period results in the lifespan extension in worms[48]. We did not address the timing issue in the current study and thus when and how Rubicon functions in neurons have to be clarified in future study.

Recent studies have revealed a positive correlation between activation of autophagy and longevity. However, the fundamental question of why autophagic activity declines with age in many organisms remains unanswered. The results of this study show, for the first time, that increase of Rubicon during aging could help explain age-dependent impairment of autophagy, thereby curtailing animal lifespan and promoting age-associated phenotypes. How Rubicon is increased with age is of particular interest.

We found that Rubicon levels were at least altered by several lifespan-extending conditions in worm and mouse tissues. These observations could help study the underlying mechanism. In the recent study, Prof. Levine's group demonstrates that the disruption of Beclin1-BCL2 extends mouse lifespan and healthspan[12]. As mentioned above, Rubicon is originally identified as Beclin 1-binding protein. While BCL-2 inhibits Beclin1 complex involved in the initiation of autophagy, another Beclin 1 complex containing Rubicon involved autophagosome-lysosome fusion processes. How different Beclin1 complexes are regulated during animal aging is particularly interested and further studies focusing on Rubicon and class III PI3K complex will open up novel paths to extending animal lifespan.

## Methods

***C. elegans* growth conditions and strains**. Nematodes were cultured using standard techniques at 20 °C on nematode growth medium (NGM) agar plates with *E. coli* strain OP50, unless otherwise noted. The following worm strains were used in the study; N2(WT), DA2123, *adIs2122 [lgg-1p::GFP::lgg-1 + rol-6(su1006)]*; AM140, *rmIs132 [unc-54p::Q35::YFP]*; TU3401, *sid-1(pk3321) V; uIs69 [pCFJ90 (myo-2p::mCherry) + unc-119p::sid-1]*; VP303, *rde-1(ne219) V; kbIs7 [nhx-2p::rde-1 + rol-6(su1006)]*. NR350, *rde-1(ne219) V; kzIs20 [hlh-1p::rde-1 + sur-5p::NLS::GFP]*; NR222, *rde-1(ne219) V; kzIs9 [(pKK1260) lin-26p::NLS::GFP + (pKK1253) lin-26p::rde-1 + rol-6(su1006)]*; CB1370, *daf-2(e1370)III*; DA465, *eat-2(ad465) II*; CB4037, *glp-1(e2141ts)III*; MQ887, *isp-1(qm150)IV*; MAH215, *sqIs11 [lgg-1p:: mCherry::GFP::lgg-1 + rol-6]*; MAH44, *glp-1(e2141ts) III*; adIs2122 *[lgg-1p::GFP:: lgg-1 + rol-6(su1006)]*; MAH14, *daf-2(e1370) III*; adIs2122 *[lgg-1p::GFP::lgg-1 + rol-6(su1006)]*; *eat-2(ad465)*; adIs2122*[lgg-1p::GFP::lgg-1 + rol-6(su1006)]*. MAH242, *sqIs24 [rgef-1p::GFP::lgg-1 + unc-122p::RFP]* was crossed with TU3401 to generate the neuron specific sensitive strain expressing neuron specific GFP::LGG-1. *hTFR:: GFP* is a kind gift from Prof. Grant (Rutgers University)[49]. Strains used in this study were summarised in Supplementary Table 2.

**Plasmid construction and transgenesis**. For *rub-1:EGFP* translational fusion constructs, *rub-1* 4 kb endogenous promoter plus coding sequence were cloned into pPD95.75 vector which contain EGFP tag Microinjection of the construct was carried with the co-injection marker, *myo-2p::mCherry* to generate *rub-1::EGFP*.

**Mouse**. CAG-Cre mice and Nestin-Cre mice were imported from Jackson Laboratory and Dr. Jun-ichi Miyazaki's laboratory (Osaka University), respectively. CAG-Cre mice were crossed with *Rubicon*$^{flox}$ mice[17] to produce mice with systemic Rubicon deletion. Resultant mice with the *Rubicon*- allele were backcrossed into the C57BL/6 J wild-type strain five times, followed by intercrossing between *Rubicon*$^{+/−}$ mice to generate *Rubicon*$^{−/−}$ mice and wild-type controls. The *Nestin-Cre* mice were crossed with *Rubicon*$^{flox}$ mice to produce mice harbouring homozygous deletion of Rubicon specifically in the brain. All mice used in this study were maintained on a C57BL/6 J background, with the exception of the calorie restricted mice. For those experiments, we used an adult-onset 40% calorie restriction protocol developed by Turturro et al[50]. Female BDF1 mice were reared individually in cages. The CR protocol was continued until 9 months of age. Water was provided ad libitum. Sequence information of primer pairs used for genotyping is available upon request. Experimental procedures using mice were approved by the Institutional Committee of Osaka University. All relevant ethical guidelines were complied with. All mice were maintained under specific pathogen-free conditions and

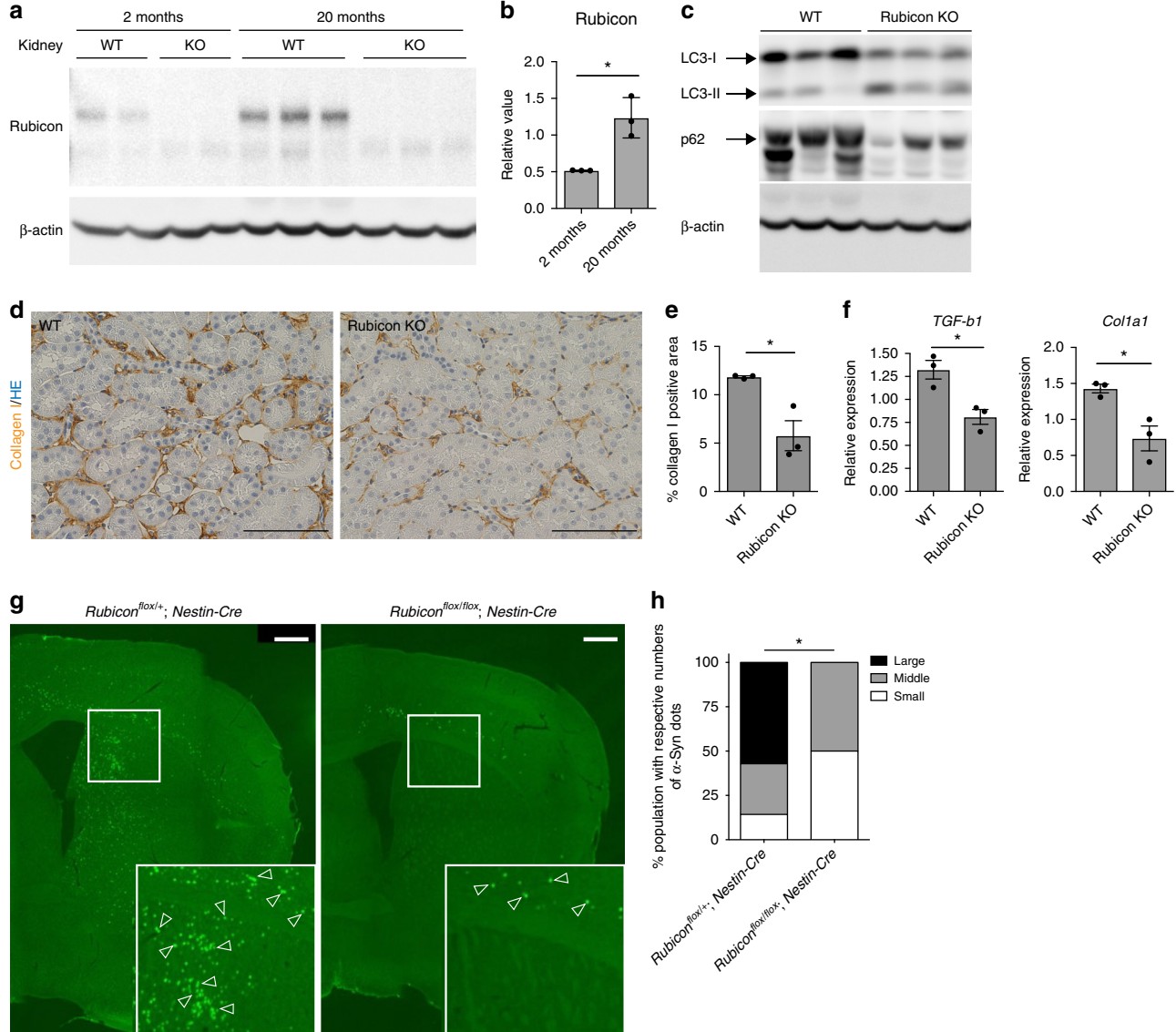

**Fig. 5** Knockout of Rubicon ameliorates age-associated phenotypes in mice. **a** Rubicon protein level in kidney was elevated in 20-month-old mice relative to 2-month-old mice. Western Blot from WT mice and Rubicon KO (KO) mice shows the specific bands of Rubicon. **b** A quantification of Western Blot shown in **a**. Value represent means ± s.e.m (2 months, $n = 3$; 20 months, $n = 3$). P value (*$P < 0.05$) was determined by $t$-test. **c** Western blotting samples showing LC-3 and p62 in mouse kidney at 20 months of age. **d** Immunohistochemical images of 20-month-old kidney showing that collagen I–positive area was reduced in Rubicon-knockout (Rubicon KO) kidney relative to WT. P value (*$P < 0.05$) was determined by $t$-test. **e** A quantification of collagen I–positive area in **d**. Values represent means ± s.e.m (WT, $n = 5$; Rubicon KO, $n = 6$). P value (*$P < 0.05$) was determined by $t$-test. **f** qRT-PCR analysis of several fibrosis markers (*TGF-b1, Col1a1*) in 20-month-old kidney. Values represent means ± s.e.m (WT, $n = 5$; Rubicon KO, $n = 6$). P value (*$P < 0.05$) was determined by $t$-test. **g** Lewy body and Lewy neurite–like inclusions containing phosphorylated α-Syn (green punctae, indicated by open arrowheads) were less abundant in neuronal Rubicon-knockout (*Rubicon$^{flox/flox}$: Nestin-Cre*) mice than in controls (*Rubicon$^{flox/+}$: Nestin-Cre*), 10 months after the injection. **h** Percentages of mice having small (<100), moderate (100–200), or large (>200) numbers of phosphorylated α-Syn positive signals. P-value (*$P < 0.05$) was determined by $\chi^2$ test (*Rubicon$^{flox/+}$: Nestin-Cre*, $n = 7$; *Rubicon$^{flox/flox}$: Nestin-Cre*, $n = 6$). Scale bars, 100 μm (**d**); 500 μm (**g**)

treated with humane care under approval from the Animal Care and Use Committee of Osaka University Medical School (Osaka, Japan).

**Fly stocks and culture conditions**. Flies were grown on standard cornmeal–agar–yeast-based medium at 25 °C. Transgenic fly lines bearing *UAS-MJDtrQ27* (#8149), *UAS-MJDtrQ78s* (#8150), *UAS-MJDtr-Q78w* (#8141), *UAS-GFP-IR* (#9330), *UAS-dRubicon (CG12772)-IR* (#43276), *UAS-GFP-mCherry-Atg8a* (#37749), *da-GAL4* (#55849) and *elav-GAL4*[c155] (#458) were obtained from the Bloomington Stock Center. Transgenic fly lines harbouring *GMR-GAL4* were described previously[51].

**Lifespan analysis**. For worms, synchronised eggs were obtained from 4–6 h egg lays on RNAi plates. Starting with day 1 adults, lifespan experiments were set up at a density of 15–20 animals per plate, and carried out at 20 °C. Worms were transferred to new plates every other day. Survivorship was also counted every other day. In case of *glp-1* background, the egg lay was conducted at 15 °C and shifted to 25 °C to induce germline-less phenotype for 2days. The lifespan experiment of *glp-1* animals was carried out at 20 °C from day1 adult stage. All of RNAi lifespan was carried out from egg onward. Death was scored as the absence of any movement after stimulation with a platinum wire. Worms that underwent internal hatching or bursting vulva, or that crawled off the plates, were censored. For TOR RNAi experiments, knockdown was performed from adult onward. For statistics, we used in house excel data sheet that can perform log-rank test[52]. Experimental flies were raised at a standard medium, allowed to mate for 48 h after emerging, then sorted using $CO_2$ anaesthesia. Vials were changed without anaesthesia to fresh food every 2–3 days, and deaths were scored until all flies were dead. More than one hundred animals per each experimental group were used in the

analyses. All lifespan experiments including repeats in worms are shown in Supplementary Data 1.

**RNA interference**. RNA interference (RNAi) was conducted by feeding HT115 (DE3) bacteria transformed with vector L4440, which produces dsRNA against the targeted gene. Synchronised eggs were placed on the indicated RNAi plates containing IPTG and ampicillin. RNAi clones were obtained from the Ahringer or Vidal RNAi library. The *let-363/TOR* RNAi clone was a gift from Dr. Hansen (Sanford-Burnham Medical Research Institute). For double knockdown experiments, equal volumes of bacteria for respective RNAi are mixed and seeded on plates. Luciferase (L4440::Luc) RNAi were used as non-targeting controls. In case of *glp-1* background, the egg lay was conducted at 15 °C and shifted to 25 °C to induce germline-less phenotype for 2days. For all RNAi experiments, except for TOR RNAi lifespan, RNAi knockdown was conducted from egg onward (whole-life knockdown).

**RNA extraction and qRT-PCR**. Worms, flies and mouse tissue samples were harvested in TRIzol (Invitrogen) or QIAZOL (QIAGEN) at respective time points. For worms, about 200 worms per condition were collected. Total RNA was extracted using RNeasy kit (QIAGEN). cDNA was generated using iScript (Bio-Rad) or PrimeScript RT reagent kit (Takara). qRT-PCR was performed with *Power* SYBR Green (Applied Biosystems) on a ViiA 7 Real-Time PCR System (Applied Biosystems) or with the KAPA SYBR Fast qPCR kit (KAPA Biosystems) on a CFX96 Real-Time PCR Detection System (Bio-Rad). Four technical replicates were performed for each reaction. *ama-1* or *cdc-42* (worms), *Hsc70* (flies), *18 s rRNA* (mice), and *GAPDH* (human) were used as internal controls. Sequences of qRT-PCR primers are shown in Supplementary Table 1.

**Autophagic flux assay in worms**. Autophagy flux assay in *C. elegans* was performed as described previously[1]. Briefly, After control or *rub-1* RNAi knockdown, 50 μM of BafA in 0.005% DMSO or 0.005% DMSO only was co-injected with 2.5 μg/ml Cascade Blue Dextran into the intestine or head region of day1 adult worms. For whole body knockdown, DA2123 (GFP::LGG-1 transgenic animals) were used, while for neuron specific knockdown newly generated strains created by crossing MAH242 with TU3401 were used. After the injection, animals were allowed to recover on respective RNAi plates for two hours. Recovered worms were then anaesthetised by 0.1% NaN₃ and mounted on 2% agar pad. GFP::LGG-1 fluorescence was captured using a FV3000 confocal microscopy (Olympus). Six to ten worms were imaged for each condition and the experiments were repeated three times. No. of GFP::LGG-1 punctae of DMSO injected and BafA injected worms in the anterior intestines and neuros were quantified using Image J.

**Multi-worm tracking analysis**. A 13 cm × 10 cm agar-filled plate was divided into four regions of equal area. To keep animals from moving between regions, the regions were surrounded with glycerol, an aversive stimulus for *C. elegans*. Worms from a given experimental group were placed in one of the four regions, and multiple experimental groups were tested simultaneously. An adapted version of the Multi-Worm Tracker[26] was used to record the locomotion of *C. elegans* on the agar plates. Locomotion was recorded for 10 min and the recording was analysed using Choreography (part of the Multi-Worm Tracker software) and custom-written scripts to organise and summarise the data. Animal tracks were collected as time series of centroid position for each frame of the final 2 min of the recording. The initial 8 min were ignored to allow animal recognition by the tracker to stabilise. The following Choreography filters were used to avoid image artefacts: –shadowless and -t 10. The speed of an individual was calculated as the sum of distances between sequential centroids, divided by the duration of the track. Experimental groups were summarised using mean and standard error of the mean, weighted by the duration of an animal's track.

**dRubicon antibody**. To generate an affinity purified antibody against dRubicon, C + EVPEEVHEKLQQAS peptide corresponding to amino acids 317–330 of dRubicon was used to immunise rabbits. The antibody was purified from the antiserum by using antigen peptide–conjugated resin. The peptides and antibody were prepared by Eurofins Genomics Inc. (Japan).

**Climbing assay**. The climbing assay was performed according to a published protocol[53] with slight modifications. Ten to twenty flies were placed in a conical glass tube (length, 15 cm; diameter, 2.5 cm) without anaesthesia. Ten seconds after tapping the flies to the bottom of the tube, the numbers of flies in each vertical area were counted and scored as follows: score 0 (0–1.9 cm), 1 (2–3.9 cm), 2 (4–5.9 cm), 3 (6–7.9 cm), 4 (8–9.9 cm), 5 (10–15 cm). Three trials were performed on each group at 20 s intervals, and the climbing score was calculated as follows: each score multiplied by the number of flies was divided by the total number of flies, and the mean score of the 3 trials was calculated. Results are presented as the mean ± S.E. of the scores obtained in 3–9 independent experiments.

**Microscopy and quantification**. To monitor autophagic activity, GFP::LGG-1 and mCherry::GFP::LGG-1 animals at the L4 stage or day1 adult stage were

anaesthetised in 0.1% sodium azide, and images were acquired using an Olympus FV1000 or FV3000 confocal microscope. Using these images, GFP and/or mCherry puncta in the anterior intestinal region or in the pharyngeal region were counted and quantified. In each experiment, images from six to ten worms per condition were captured and the experiments were repeated three times. For Rub-1::EGFP and Q35::YFP worms, fluorescent images of the aligned whole worms on ice cold empty plates were captured by stereomicroscope SZX16 equipped with DP80 CCD camera (Olympus) and the number of polyQ aggregates per worms was counted.

**Fly eye imaging**. Light microscopic images of 1-day-old adult female flies were taken using a stereoscopic microscope model SZX16 (Olympus) with a CCD camera (DP22, Olympus). Numbers of necrotic patches per compound eye were counted for each genotype.

**Autophagic flux assay in *Drosophila***. To match the feeding status, 4-day-old flies expressing GFP-mCherry-Atg8a either with or without dRubicon-IR under the da-GAL4 driver were starved by 0.75% agar for 15 h and refed for 4 h. The brains were dissected, fixed in 4% formalin in PBS, incubated with 80% glycerol overnight, and then mounted with DAPI fluoromount-G (SouthernBiotech). Fluorescent images of KC layer were taken by confocal microscope (Leica TCS SP8) and the number of GFP and/or mCherry puncta in 25 × 50 μm ROI containing 150–200 cells were counted. Three un-overlapping images were used for quantification of each KC layers.

**Stress resistance assay**. Day1 adult worms were subjected to oxidative stress (4.4 mM H₂O₂ in the empty plate) or heat stress (35 °C) for 3 h or 7 h, respectively and the survived worms were counted. 30 worms were used per condition and the experiments were repeated three times.

**Western blotting**. Worms or mouse tissues were lysed in lysis buffer (50 mM Tris/HCl [pH 7.4], 150 mM NaCl, 1 mM EDTA, 0.1% NP-40, protease and phosphatase inhibitor cocktail [Roche]) using a homogeniser. After centrifugation, the resultant supernatants were subjected to protein quantification and western blotting. Protein lysates of worm and mouse tissues were separated by SDS-PAGE and transferred to PVDF membranes, which were then blocked and incubated with specific primary antibodies. Primary antibodies and dilutions used for mouse tissues western blotting were as follows: Rubicon (Cell Signalling Technology, #8465, 1:500), LC3 (Cell Signalling Technology, #2755, 1:1000), p62 (MBL, PM045, 1:1000), and β-actin (Sigma Aldrich, A5316, 1:8000). In fly, to assess the total amount of dRubicon, two fly bodies were lysed in 100 μl of 2 × SDS buffer (2% SDS, 125 mM Tris-HCl, pH 6.8, 4% SDS, 20% Glycerol, 0.01% bromophenol blue, and 10% 2-Mercaptoethanol). The lysates were heated at 99 °C for 10 min followed by centrifugation at 15,000 × g for 10 min and the supernatants were collected. The protein lysates of 0.75 bodies per lane were separated by 5–12% polyacrylamide gels, transferred to PVDF membranes. Blocking with 5% skim milk in PBS containing 0.1% Tween 20. The antibodies used in this study were as follows: anti-dRubicon (1:20,000), anti-actin (JLA20, 1:2,000, Developmental studies Hybridoma Bank). Signal intensities were quantified by densitometry using ImageJ. Uncropped images were provided in Supplementary Fig. 10.

**Immunohistochemistry**. For flies, adult female brains (3 days after eclosion) were dissected, fixed in 4% formalin in PBS, blocked with 50% Block Ace (Dainippon Sumitomo Pharmaceuticals) in PBS/T (0.5% Triton X-100 in PBS) and incubated with an anti-HA antibody (3F10, 1:500, Sigma) to stain MJDtrQ78 protein with HA-tag. The immunostainings were visualised with an Alexa 488-conjugated anti-rat antibody and nuclei were stained with DAPI. Images of Kenyon cell layer (KC layer) were taken by confocal microscopy (LSM780, Zeiss). Signal intensities of HA staining were quantified using Image J 1.51 s software. Ten bilateral KC layers from five animals were analysed for each group. For mice, immunohistochemical staining for collagen I was performed on paraffin-embedded sections. After antigen retrieval by autoclaving in 0.01 mmol/L citrate buffer (pH 6.0) for 10 min at 120 °C, the sections were blocked with 3% BSA in PBS for 60 min. The blocked sections were incubated with primary antibodies, anti-Collagen I (abcam, ab34710, 1:400) at 4 °C overnight, incubated for 30 min at room temperature, followed by detection using a HRP-diaminobenzidine compound (Nichirei Corp.). Sections were counterstained with hematoxylin.

**Injection and detection of recombinant α-Syn Fibrils**. Injection of fibrils and their detection were described previously[38,39]. In brief, prepared fibrils were diluted in sterile PBS and sonicated before intracerebral injection. Mice between 8 and 10 weeks of age were anaesthetised with chloral hydrate (250 mg/kg, i.p.) and stereotaxically injected in both hemispheres (co-ordinates: + 0.2 mm relative to Bregma, + 2.0 mm from midline) with recombinant α-Syn fibrils (1 μl of 1 mg/ml). Animals were killed at 10 months after birth. Frozen brain sections including the fibril-injected area were obtained using a CM3050S cryostat (Leica Biosystems). Sections were incubated with mouse monoclonal antibody pSyn#64 (dilution, 1:1000, Wako Pure Chemical Industries), followed by Alexa Fluor 488–conjugated

secondary antibodies (dilution, 1:1000, Invitrogen) for detection. Images were acquired and quantified on a Biorevo BZ-9000 (Keyence).

**Statistical analysis**. Results are presented as means ± s.e.m. Statistical tests were performed with either one- or two- way ANOVA with Tukey's test or *t*-test using GraphPad Prism (GraphPad Software) or Excel (Microsoft Office 2011).

**Reporting summary**. Further information on experimental design is available in the Nature Research Reporting Summary linked to this article.

## Data availability

All relevant data are available within the manuscript and its supplementary information or from the authors upon reasonable request.

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

## Acknowledgements

We thank Prof. Dr. Hansen and Dr. Kumsta (Sanford-Burnham Medical Research Institute) for *let-363/TOR* RNAi clone, bafilomycin injection protocol and MAH215 strain. We thank Prof. Grant for providing hTFR::GFP transgenic worms. Many thanks to *C. elegans* Genetic Center (CGC) at the University of Minnesota. S.N. is supported by AMED-PRIME (17gm6110003h0001), JSPS KAKENHI, the Senri Life Science Foundation, the Takeda Science Foundation, the Nakajima Foundation, and the MSD Life Science Foundation. M.S. is supported by JSPS KAKENHI and the Ichiro Kanehara Foundation. T.Y. is supported by AMED under grant number JP17gm5010001 and JP17gm0610005, MEXT/JSPS KAKENHI, JST CREST (JPMJCR17H6) and HFSP grant.

## Author contributions

S.N. M.O., M.S. and T.Yoshimori designed the study. S.N. A.Tokumura and S.K. conducted worm experiments. S.N. and K.I. conducted multi-worm tracking analysis. A. Takahashi, M.F., T. Yamamuro, K.A., S.M., T.K., Y.M., Y. W., Y.T., H.M., Y.I. and T. Yoshimori designed and performed mouse experiments. N.T. and T.S.K. generated calorie restriction mouse. M.H., K.Y., and Y.O. helped data analysis. M.O., M.S., K.F. and K.S. designed and conducted *Drosophila* experiments. A.A. helped worm experimental design. S.N. M.O., M.S. and T.Yoshimori wrote the manuscript.

## Additional information

**Competing interests:** The authors declare no competing interests.

**Journal Peer Review Information**: *Nature Communications* thanks the anonymous reviewers for their contribution to the peer review of this work. Peer reviewer reports are available.

