## [Peer Review File · Nature Communications]

Editorial Note (1): This manuscript has been previously reviewed at another journal that is not operating a transparent peer review scheme. This document only contains reviewer comments and rebuttal letters for versions considered at Nature Communications.

Editorial Note (2): Parts of this Peer Review File have been redacted as indicated to maintain the confidentiality of unpublished data.

Reviewers' comments:

Reviewer #1 (Remarks to the Author):

I feel that I should work as an arbitrary referee, since the paper has been already reviewed. Overall, the observations are interesting. To my taste, authors have properly responded to most of the reviewers' suggestions. However there are some points that were suggested but not examined.

- Reviewer#1 asked for a setup, where Rubicon overexpression reverses the longevity phenotype in *daf-2*, *eat-2* mutants. The authors argued that CeRubicon knockdown does not further extend longevity of *eat-2*, *daf-2*, *glp-1* and *isp-1*. However it would also be interesting to look at the autophagy levels in this conditions too.

- Reviewer#2 "The authors go on to suggest that several long-lived *C. elegans* mutants, which are shown to display reduced levels of Rubicon, require reduced levels of Rubicon for their longevity without supporting this claim."
The explanation that downregulation of CeRubicon does not extend *daf-2*, *isp-1*, *eat-2* and *glp-1* lifespan, is not the right answer for this comment. Only overexpression of CeRubicon in the long-lived *C. elegans* mutants could support this claim (also Reviewer#1 is asking for overexpression experiments in long-lived mutants!). This maybe a very intricate experiment, therefore the authors may solve this problem by deleting the claim.

- As reviewer#2 suggested, also mRNA levels in mice should be tested in mice as well. Moreover, qPCR is missing in the CeRubicon/*atg-18* experiment as reviewer#2 mentioned.

- During the review process a climbing experiment was made with 3 groups (*elav_control*, *elav_MJDtrQ78w*, *elav_MJDtrQ78w/Rubicon-IR*) . Why is the *elav_Rubicon-IR* group missing? Also qPCR of all groups should be made, to exclude that the effect comes from different expression of *MJDtrQ78*.

- The authors showed that the autophagic flux is significantly increased in flies with a systemic knockdown of dRubicon. Are the autophagy levels also increased in the systemic knockdown of dRubicon in *Drosophila*? (Westernblot with Gabarap or p62)

- The figures are confusing, because mice, *Drosophila* and *C. elegans* data is mixed up. Please sort the figures in a more logical way. They should be understandable at a glance.

Reviewer #2 (Remarks to the Author):

In the manuscript "Suppression of autophagic activity by Rubicon is a signature of aging" by Nakamura et al., the authors aim to answer the question why autophagic activity declines with

age. They suggest that the expression levels of Rubicon, a negative regulator of autophagy increases with age in a conserved manner and that this age-dependent increase limits longevity and several health parameters. The authors argue that this amelioration of age-associated phenotypes is dependent on active autophagy, but fail to convincingly show the autophagy dependence.

In this slightly revised manuscript (from the previous Nature version), the authors have only minimally addressed the reviewers' concerns, and the main issues still stand (with new additional ones) as follows; proper controls are missing, especially in regards to autophagy measures, model systems should be used more consistently, and the findings need to be put in better context and not overstated. Overall, the manuscript needs significant changes and several experiments will have to be included before this manuscript can be considered for possible publication in Nature Communications.

Specific comments:

A) Expression of Rubicon with age

1. The expression levels of Rubicon with age should be assessed in more than two time points in *C. elegans*.
2. The expression of Rubicon in *Drosophila* tissues with age should be included, especially in the tissues tested, i.e., eyes and muscles.
3. mRNA levels of Rubicon with age should be analyzed in mice.

B) Autophagy assays

1. Autophagy flux assays

The authors now use an mCherry::GFP::LGG-1 tandem reporter for autophagy measures. This reporter can be used to determine the pool sizes of autophagosomes and autolysosomes at steady-state level, but does not by itself report on autophagy flux. To perform autophagy flux experiments, autophagy inhibitors must be included, for instance treatment with BafA. The quantification of autolysosomes in *C. elegans* intestine seems very different from published data (Chang et al., *Elife* 2017).

2. Tissue-specific autophagy measurements

The authors need to include autophagy measurements in neurons. Multiple neuronal GFP::LGG-1 and tandem reporters are available and have been used for the quantification of autophagic structures pan-neuronally or in select neuronal types. These results are critical to support the relevance of neuronal autophagy (potentially downregulated by Rubicon) in lifespan extension.

3. Autophagy measurements after epistasis experiments

The authors show that the reduction of *bec-1* and Rubicon ameliorates the increase of autophagic punctae induced by Rubicon knockdown. Does Rubicon interact with *BEC-1* as reported in other species? Knockdown of other autophagy genes that are not putative Rubicon interactors should be performed. This experiment needs to include a control of knockdown of the specific genes. In general, it is not clear where the authors performed parallel RTPCR experiments, e.g. for double-RNAi experiments in neurons? (Supp. Fig 3d?)

4. Ideally, autophagic activity should be measured over several time points.

5. While the authors now cite Wilhelm et al (*Genes and Development* 2017), they fail to comment on the relevant data (namely that inhibition of at least *bec-1* in neurons of young animals does not affect lifespan; note that these data are in Supplements).

6. While the inclusion of new *Drosophila* lifespan data are appreciated, autophagy measurements in *Drosophila* should be included for completion.

C) Rubicon overexpression strain

The authors have included novel data with a newly generated overexpression strain (were multiple independent strains generated and analyzed?). The images of the Rubicon overexpression are very poor and no conclusions can be drawn on expression pattern. The Day 7 images could be autofluorescence. Is Rubicon indeed expressed in neurons?

Lifespan experiments of the Rubicon overexpression in long-lived mutants is necessary, if the authors would like to claim that the "longevity of these animals is partly conferred by the reduction of rubicon expression". Generally, this new reagent is really underused in several contexts (including in protein aggregation).

D) mTOR claims

As noted in earlier review, mTOR epistasis experiment and conclusion is still inherently flawed (as none of the conditions are null conditions). Suppl. Fig 4g claims to have analyzed P-S6K levels, but with no size marker (or, ideally, a positive control or the used antibody), this claim can essentially not be evaluated. While RTPCR experiments were done to check effects of mTOR knockdown, how come protein levels were not assayed in the new tagged strain? As noted above, the authors need to consistently measure outputs.

E) Mondo claims

As noted in earlier review, the authors discuss Mondo as a potential upstream transcriptional regulator of Rubicon, but this objective does not seem well justified in the context of the manuscript otherwise being focused on Rubicon levels over time.

F) Endocytosis

Authors have not addressed whether endocytosis, in addition to autophagy, could be important for Rubicon's functions in invertebrates. This possibility should at the very least be discussed in the text.

G) Text and references

Resubmitted manuscript has not been formatted for Nature Communications from Nature version, i.e., no formal Results or Discussion sections (would also have been helpful if it was paginated). All figure legends should clearly list number of repeats (as is, only some of them do). For completion, the authors should cite longevity studies of Atg1 overexpression in fly.

Reviewers' comments:

Reviewer #1 (Remarks to the Author):

I feel that I should work as an arbitrary referee, since the paper has been already reviewed. Overall, the observations are interesting. To my taste, authors have properly responded to most of the reviewers' suggestions. However there are some points that were suggested but not examined.

Thank you so much for being interested in our work. We have addressed raised questions in the following section.

- Reviewer#1 asked for a setup, where Rubicon overexpression reverses the longevity phenotype in *daf-2*, *eat-2* mutants. The authors argued that CeRubicon knockdown does not further extend longevity of *eat-2*, *daf-2*, *glp-1* and *isp-1*. However it would also be interesting to look at the autophagy levels in this conditions too.

Thank you so much for the valuable comments. Now we checked the numbers of GFP::LGG-1 puncta in several long lived animals (*daf-2*, *glp-1* and *eat-2*) after knockdown using control or CeRubicon RNAi. We found that knockdown of CeRubicon did not further increase the numbers of puncta in these long-lived animals. These results are now shown in Supplementary Fig. 5. Although we tried the cross twice, we could not create GFP::LGG-1 in *isp-1* background. After the cross, GFP::LGG-1 expression pattern was largely disrupted. Thus we could not properly assessed GFP::LGG-1 puncta in *isp-1* background.

- Reviewer#2 "The authors go on to suggest that several long-lived *C. elegans* mutants, which are shown to display reduced levels of Rubicon, require reduced levels of Rubicon for their longevity without supporting this claim."

The explanation that downregulation of CeRubicon does not extend *daf-2*, *isp-1*, *eat-2* and *glp-1* lifespan, is not the right answer for this comment. Only overexpression of CeRubicon in the long-lived *C. elegans* mutants could support this claim (also Reviewer#1 is asking for overexpression experiments in long-lived mutants!). This maybe a very intricate experiment, therefore the authors may solve this problem by deleting the claim.

We really appreciate this suggestion and I agree that the even overexpression experiment can not fully address the claim. Therefore, as reviewer suggested, we simply removed the claim that reduction of CeRubicon is part of several longevity mechanism. Instead, we simply mentioned

that genetic interaction between CeRubicon and other longevity pathway. Now these results are shown in Supplementary Fig. 4 c-g.

- As reviewer#2 suggested, also mRNA levels in mice should be tested in mice as well. Moreover, qPCR is missing in the CeRubicon/atg-18 experiment as reviewer#2 mentioned.

Thank you for the comments. We have now conducted these qPCR experiments. We found that mouse Rubicon is increased at transcript level with age in kidney and liver. The results are shown in Supplementary Fig. 9a, b. qPCR for CeRubicon/atg-18 is also shown in Supplementary Fig. 2b and Supplementary Fig. 7b.

- During the review process a climbing experiment was made with 3 groups (elav_control, elav_MJDtrQ78w, elav_MJDtrQ78w/Rubicon-IR) . Why is the elav_Rubicon-IR group missing? Also qPCR of all groups should be made, to exclude that the effect comes from different expression of MJDtrQ78.

We have now added elev_Rubicon-IR group in the climbing assay (Fig. 4i). We have also conducted qPCR for MJDtrQ78 and confirmed that expression of MJDtrQ78 did not change among the groups (Supplementary Fig. 8h). Thank you.

- The authors showed that the autophagic flux is significantly increased in flies with a systemic knockdown of dRubicon. Are the autophagy levels also increased in the systemic knockdown of dRubicon in drosophila? (Westernblot with Gabarap or p62)

We have checked Atg8 protein by Westernblot using whole body or neuron specific dRubicon knockdown samples. However, in both cases, we could not observe drastic changes of Atg8 II form compared to control (please see the figure below). This could be due to the heterogeneous population of knockdown cells. At this point, we think tfAtg8 results shown in Supplementary Fig. 8 e, f, seem to be the best way to address autophagic activity, like other drosophila researchers do (DeVorkin et al., JCB 2014).

- The figures are confusing, because mice, drosophila and C. elegans data is mixed up. Please sort the figures in a more logical way. They should be understandable at a glance.

We are very sorry for the confusing structure. We have now tried to organize figures to summarize by each species and we hope the revised version of manuscript becomes easy to follow and understand.

Reviewer #2 (Remarks to the Author):

In the manuscript “Suppression of autophagic activity by Rubicon is a signature of aging” by Nakamura et al., the authors aim to answer the question why autophagic activity declines with age. They suggest that the expression levels of Rubicon, a negative regulator of autophagy increases with age in a conserved manner and that this age-dependent increase limits longevity and several health parameters. The authors argue that this amelioration of age-associated phenotypes is dependent on active autophagy, but fail to convincingly show the autophagy

dependence.

In this slightly revised manuscript (from the previous Nature version), the authors have only minimally addressed the reviewers' concerns, and the main issues still stand (with new additional ones) as follows; proper controls are missing, especially in regards to autophagy measures, model systems should be used more consistently, and the findings need to be put in better context and not overstated. Overall, the manuscript needs significant changes and several experiments will have to be included before this manuscript can be considered for possible publication in Nature Communications.

First of all, we would like to thank the reviewer for valuable and constructive comments. We answered to the specific comments in the following section.

Specific comments:

A) Expression of Rubicon with age

1. The expression levels of Rubicon with age should be assessed in more than two time points in *C. elegans*.

Thank you for the suggestions. We have collected time course samples at day1, 3, 5, 7 and conducted qPCR. We found that CeRubicon is upregulated from day3 onward. These results are shown in Fig. 1e.

2. The expression of Rubicon in *Drosophila* tissues with age should be included, especially in the tissues tested, i.e., eyes and muscles.

We have developed antibody against dRubicon and checked the expression of dRubicon over time. Similar to worm and mouse tissues, we could observe dRubicon is upregulated in whole body with age. The results are now shown in Fig. 4a, b.

3. mRNA levels of Rubicon with age should be analyzed in mice.

Thank you for the suggestion. We have now conducted qPCR analysis to check if mouse Rubicon is increased at transcript level. Indeed, similar to worms, Rubicon transcript levels are increased both in kidney and liver with age and these results are shown in Supplementary Fig.9 a, b.

B) Autophagy assays

1. Autophagy flux assays

The authors now use an mCherry::GFP::LGG-1 tandem reporter for autophagy measures. This reporter can be used to determine the pool sizes of autophagosomes and autolysosomes at steady-state level, but does not by itself report on autophagy flux. To perform autophagy flux experiments, autophagy inhibitors must be included, for instance treatment with BafA. The quantification of autolysosomes in *C. elegans* intestine seems very different from published data (Chang et al., *Elife* 2017).

I appreciated the valuable comments. We conducted bafilomycin A1 injection experiments to measure the autophagy flux. Indeed, we confirmed that whole body knockdown of CeRubicon increased autophagy flux. These results are shown in Fig. 1c and Supplementary Fig. 2e. We don't know the reason for the difference between our results and their results in terms of autolysosomes. We could observe few autolysosomes in intestine at steady state levels. One possibility is the difference of generations used for each experiment, but honestly we don't have clear answer for this.

2. Tissue-specific autophagy measurements

The authors need to include autophagy measurements in neurons. Multiple neuronal GFP::LGG-1 and tandem reporters are available and have been used for the quantification of autophagic structures pan-neuronally or in select neuronal types. These results are critical to support the relevance of neuronal autophagy (potentially downregulated by Rubicon) in lifespan extension. We obtained neuron specific GFP::LGG-1 strain MAH242(sqIs24 [rgef-1p::GFP::lgg-1 + unc-122p::RFP]) and crossed this with neuron specific RNAi sensitive strain TU3401. By using this, we conducted bafilomycin A microinjection experiments after knockdown of control/CeRubicon and measured the autophagic flux. We confirmed the autophagic flux is increased in neuron by neuron specific CeRubicon knockdown. These results are shown in Fig. 3 e, f. Thank you.

3. Autophagy measurements after epistasis experiments

The authors show that the reduction of bec-1 and Rubicon ameliorates the increase of autophagic punctae induced by Rubicon knockdown. Does Rubicon interact with BEC-1 as reported in other species? Knockdown of other autophagy genes that are not putative Rubicon interactors should be performed. This experiment needs to include a control of knockdown of the specific genes. In general, it is not clear where the authors performed parallel RTPCR experiments, e.g. for double-

RNAi experiments in neurons? (Supp. Fig 3d?)

Thank you for the valuable comments. We developed antibody against CeRubicon but the serum did not detect CeRubicon properly. Thus we failed to detect the interaction of bec-1 and CeRubicon in vivo. Instead, we cloned bec-1, CeRubicon and *C. elegans* Plekhh1 homolog (CePlekhh1) and transfected them in HEK293 cells. We found that bec-1::FLAG is co-immunoprecipitated with CeRubicon::GFP rather than CePlekhh1::GFP (please see the following picture, red arrow indicates CeRubicon-GFP). However, we don't know how much this result reflects in vivo situation. Therefore, we only show this result in this response file. We now added the results of GFP::LGG-1 count after the concomitant knockdown of atg-18 (Supplementary Fig.1c). qPCR results in TU3401 (neuron specific RNAi strain) are also shown (now in Supplementary Fig. 7b).

4. Ideally, autophagic activity should be measured over several time points.

We would like to address which timing of basal autophagic activity is essential for longevity in our future work. Therefore, we would like to provide these time course data together with detailed lifespan experiments in our upcoming study. We hope the revised version of our manuscript have already provided sufficient sets of data to support our claim. Thank you.

5. While the authors now cite Wilhelm et al (Genes and Development 2017), they fail to comment on the relevant data (namely that inhibition of at least bec-1 in neurons of young animals does not affect lifespan; note that these data are in Supplements).

Thank you for the comments. We found the relevant sentence in the manuscript; “Intriguingly, neuron-specific inactivation of bec-1 and vps-34 at day 9 strongly extended MTL by up to 57% (Fig. 5A), whereas atg-7 did not alter life span, and lgg-1 significantly reduced MTL (Fig. 5A). We observed a marked reduction in life span upon neuron-specific bec-1 RNAi at day 0 (Supplemental Fig. S9A), which indicates a conserved neuronal AP effect for bec-1”. We now commented about the discrepancy in the discussion section.

6. While the inclusion of new Drosophila lifespan data are appreciated, autophagy measurements in Drosophila should be included for completion.

We have measured autophagy activity using tfAtg8 which is well established method in fly and autophagic activity is increased by dRubicon knockdown. The results are shown in Supplementary Fig. 8e, f.

C) Rubicon overexpression strain

The authors have included novel data with a newly generated overexpression strain (were multiple independent strains generated and analyzed?). The images of the Rubicon overexpression are very poor and no conclusions can be drawn on expression pattern. The Day 7 images could be autofluorescence. Is Rubicon indeed expressed in neurons?

Lifespan experiments of the Rubicon overexpression in long-lived mutants is necessary, if the authors would like to claim that the “longevity of these animals is partly conferred by the reduction of rubicon expression”. Generally, this new reagent is really underused in several contexts (including in protein aggregation).

We have provided CeRubicon::EGFP expression showing neuronal expression side by side with N2 animals to distinguish from autofluorescence (Supplementary Fig.3). Regarding overexpression, we obtained the similar suggestions from reviewer 1 and we felt that even overexpression experiment could not really address the convergent function of CeRubicon in all long-lived animals. Therefore, we simply deleted the claim and we only mentioned the genetic

interaction between CeRubicon and other longevity pathways. These are now shown in Supplementary Fig. 4c-g.

D) mTOR claims

As noted in earlier review, mTOR epistasis experiment and conclusion is still inherently flawed (as none of the conditions are null conditions). Suppl. Fig 4g claims to have analyzed P-S6K levels, but with no size marker (or, ideally, a positive control or the used antibody), this claim can essentially not be evaluated. While RTPCR experiments were done to check effects of mTOR knockdown, how come protein levels were not assayed in the new tagged strain? As noted above, the authors need to consistently measure outputs.

We have characterized the pS6 kinase antibody used in this study. We used same antibody as Mair lab used (Heinz et al., Nature, 2017), since this is the only antibody working in worms. However, during the characterization we realized the antibody react with lysate at prospective size from rsk-1 mutant which is the null mutant of S6K in worms. Therefore, we simply deleted the claim that CeRubicon knockdown did not affect TOR pathway. We only mention the genetic interaction of CeRubicon and TOR where CeRubicon knockdown is additive to TOR knockdown effect in term of lifespan regulation.

E) [redacted]

F) Endocytosis

Authors have not addressed whether endocytosis, in addition to autophagy, could be important for Rubicon's functions in invertebrates. This possibility should at the very least be discussed in the text. Thank you for this point. We obtained the hTFR (human transferrin receptor)::GFP transgenic worms from Prof Grant and checked the localization of GFP puncta after knockdown of CeRubicon. We could not observe the drastic changes by the knockdown and the results are shown in Supplementary Fig. 2f. Of course, we could not rule out the possibility that CeRubicon has a role during other endocytic pathways, CeRubicon might not be involved in recycling of the receptor like mammalian cells.

G) Text and references

Resubmitted manuscript has not been formatted for Nature Communications from Nature version, i.e., no formal Results or Discussion sections (would also have been helpful if it was paginated). All figure legends should clearly list number of repeats (as is, only some of them do). For completion, the authors should cite longevity studies of Atg1 overexpression in fly.

The manuscripts are now formatted for Nat Commun. We also tried to organize figures to summarize by each species and we hope the revised version of manuscript becomes easy to follow and understand. In the legend, no of repeats are described. Now we have cited Atg1 overexpression work (Ulgherait et al., 2014). Thank you.

Reviewers' comments:

Reviewer #1 (Remarks to the Author):

Good revision

Reviewer #2 (Remarks to the Author):

In this resubmitted manuscript, the authors have addressed many experimental issues and improved the overall read of the manuscript; however, several remaining concerns remain. These are outlined below, along with some additional suggestions for the authors to further improve their study.

Concerns:

1. Some lifespan assays are not repeated sufficiently. Standard in the *C. elegans* field for RNAi treatments is to do 3 repeats per condition/individual genes – here, two trials of multiple autophagy genes in combination with CeRubicon seems adequate (although see remaining concern about missing RTPCR tests below, but other conditions are not: CeRubicon on hypodermal-, muscle- and intestinal- RNAi strains and on long-lived mutants (one trial only). Likewise, it is not standard in the fly field to only carry out one repeat per condition. The Supplemental table with the fly data does not correctly show that male lifespans are shortened by dRubicon knockdown (i.e., minus signs should be included). The Supplemental table needs to be referenced in the figure legends so that the reader knows to go to the table for more information and repeats (not sufficient to just note this in the Methods section).
2. Although the authors tested knockdown efficiency in double RNAi with CeRubicon, they only did RTPCR for *atg-18* in double-RNAi conditions (shown in Supplemental Fig. 4b and Fig. 7b). RNAi efficiency need be tested for all genes used in double RNAi experiments.
3. The new *C. elegans* autophagy flux assays are unclear. The method section is insufficient and it is not clear what is actually measured in arbitrary units. To make claims about autophagy flux, the authors need to compare the number of autophagosomes under mock injection conditions (DMSO) and blocked autophagy (BafA) conditions. The BafA injections in wild-type animals fed control RNAi bacteria serve as a control for their BafA injections (Figure 1c and 3e). It is also unclear how old the animals were when autophagy flux was measured.
4. The images of the Rubicon overexpressor in Figure 1D have not been improved. This is critical to evaluate an increase in the expression levels of Rubicon with age. These images should be retaken (with wild-type as a control, which could be put in the supplements). The new supplemental images of Rubicon expression in the nerve-ring should also be shown with age. Importantly, these pictures do not work as a control for age-dependent increase in autofluorescence, but need to show the intestine.

Textual corrections needed:

1. The introduction should be revised and include a more thorough introduction into Rubicon (to explain what is 'unique' about Rubicon, an important part of the premise for the study).
2. The autophagy assay used in flies (Supplemental Fig. 8e, f) only reflects steady-state of autophagosome/ autolysosome pools, not flux. While such assays are not standard in the fly field, the text needs to accurately reflect the conclusions, and limitations, that can be drawn from the current experiments. A suggestion would be to move these data into the main figure, which would align all figures conceptually.
3. All figures should clearly indicate when RNAi was used and all figure legends should clearly

describe when the RNAi was initiated (whole-life or adult-only). To this point, the Methods section is missing a description of their double RNAi experiments.

4. All figures should clearly state when mRNA levels are measured versus protein levels. The nomenclature of CeRubicon is confusing, as it does not adhere to *C. elegans* nomenclature standards. To this end, the authors are encouraged to get a name formally curated in the *C. elegans* field (see Wormbase for instructions).

5. In Figure 5g, the Figure should indicate that the green punctae are alpha-syn.

6. Line 79-80: the authors mention a recent paper from the Levine lab in the discussion, it should be mentioned in the introduction, along with other studies extending lifespan by autophagy-gene modulation discussed there.

7. Line 184: The authors are not testing that increased autophagy in neurons is sufficient to extend lifespan. This needs to be rephrased to say that autophagy gene *atg-18* is required for the lifespan extending effects of neuronal Rubicon reduction.

8. Line 299: The authors should delete " while the knockdown of *bec-1*..." until the end of the paragraph.

9. Line 306. The authors do not show an "incremental" increase. This should be revised.

Suggestions:

1. Since the authors are not characterizing the CeRubicon overexpressor strains in any other assays besides lifespan, Figure 1 d,h,i could be moved to the supplement.

2. Figure 3 could be made into a neuronal-specific figure in which Figure 3b,c,d, are moved into the supplement (noting concerns about rigor/repeats listed above), and Figure S7A is moved into Figure 3.

3. The concern for claiming Mondo as a potential upstream transcriptional regulator of Rubicon still stands. The objective of investigating Mondo as a regulator of Rubicon does not seem well justified since the manuscript is otherwise focused on Rubicon levels over time. The authors are strongly encouraged to further investigate this in another manuscript, and instead combine Figure 6A,B with their other figures, now conceptually well organized by species.

4. A strain list indicating both previously published and newly generated strains would be helpful.

2nd round reviews

Responses to Reviewers

Reviewers' comments:

Reviewer #1 (Remarks to the Author):

Good revision

Thank you so much! We really appreciate your time and valuable comments so far.

Reviewer #2 (Remarks to the Author):

In this resubmitted manuscript, the authors have addressed many experimental issues and improved the overall read of the manuscript; however, several remaining concerns remain. These are outlined below, along with some additional suggestions for the authors to further improve their study.

Thank you so much for taking time. We appreciate valuable and constructive comments/suggestions for us. In the following sections, we specifically responded to each concern and comment.

Concerns:

1. Some lifespan assays are not repeated sufficiently. Standard in the *C. elegans* field for RNAi treatments is to do 3 repeats per condition/individual genes – here, two trials of multiple autophagy genes in combination with CeRubicon seems adequate (although see remaining concern about missing RTPCR tests below, but other conditions are not: CeRubicon on hypodermal-, muscle- and intestinal- RNAi strains and on long-lived mutants (one trial only). Likewise, it is not standard in the fly field to only carry out one repeat per condition. The Supplemental table with the fly data does not correctly show that male lifespans are shortened by dRubicon knockdown (i.e., minus signs should be included). The Supplemental table needs to be referenced in the figure legends so that the reader knows to go to the table for more information and repeats (not sufficient to just note this in the Methods section).

For worm, we have repeated lifespan of long-lived mutant background (*daf-2*, *isp-1*, *eat-2* and *glp-1*) and of overexpressing animals and obtained similar results. Tissue specific knockdown lifespan has been repeated 3 times. Now we included these data in supplementary Data 1. For the tissue specific knockdown lifespan experiments, we moved these data to Supplementary Figure 7 a-c based on your kind suggestion below. For fly, we have already repeated critical demographic analysis with neuron-specific dRubicon knockdown and confirmed similar tendency both in female and male (now included in Supplementary Data 1). We did not repeat whole body knockdown lifespan and it will take another 6months just for this repeat. Therefore, we moved this to the supplementary figure 8d. We hope this is reasonable. Thank you for pointing out about the expression regarding fly male lifespan. We have now inserted the minus sign to show that male lifespan is shortened by dRubicon knockdown. And thank you for your suggestions, now we mentioned about the lifespan repeats in Supplementary Data 1 in the figure legends as well.

2. Although the authors tested knockdown efficiency in double RNAi with CeRubicon, they only did RTPCR for *atg-18* in double-RNAi conditions (shown in Supplemental Fig. 4b and Fig. 7b). RNAi efficiency need be tested for all genes used in double RNAi experiments.

Thank you. We have now added our all qPCR analysis of double knockdown experiments to show the knockdown efficiency with statistical test in Supplementary Figure 3 b-d.

3. The new *C. elegans* autophagy flux assays are unclear. The method section is insufficient and it is not clear what is actually measured in arbitrary units. To make claims about autophagy flux, the authors need to compare the number of autophagosomes under mock injection conditions (DMSO) and blocked autophagy (BafA) conditions. The BafA injections in wild-type animals fed control RNAi bacteria serve as a control for their BafA injections (Figure 1c and 3e). It is also unclear how old the animals were when autophagy flux was measured.

We have now added more information including the age of animals in the method section for autophagic flux assay. Following the suggestions, we showed actual no. of GFP::LGG-1 of DMSO only injected or BafA injected worms for each RNAi condition, respectively (Figure 1c and Figure 3d). Both in intestine and in neurons, we confirmed that no. of GFP::LGG-1 was more increased

after BafA injection in *rub-1* knockdown worms compared to control knockdown, suggesting that *rub-1* knockdown increased autophagic flux. Thank you for these critical comments.

4. The images of the Rubicon overexpressor in Figure 1D have not been improved. This is critical to evaluate an increase in the expression levels of Rubicon with age. These images should be retaken (with wild-type as a control, which could be put in the supplements). The new supplemental images of Rubicon expression in the nerve-ring should also be shown with age. Importantly, these pictures do not work as a control for age-dependent increase in autofluorescence, but need to show the intestine.

Thank you so much for this critical comment. We have now retaken the representative pictures by confocal microscopy at same exposure time together with corresponding age-matched WT control (N2) to compare with autofluorescence. These pictures are now shown in Figure 1d.

Textual corrections needed:

1. The introduction should be revised and include a more thorough introduction into Rubicon (to explain what is 'unique' about Rubicon, an important part of the premise for the study).

Thank you for the suggestion. We have inserted more introduction about Rubicon including recently published paper regarding a role rubicon in NAFLD.

2. The autophagy assay used in flies (Supplemental Fig. 8e, f) only reflects steady-state of autophagosome/ autolysosome pools, not flux. While such assays are not standard in the fly field, the text needs to accurately reflect the conclusions, and limitations, that can be drawn from the current experiments. A suggestion would be to move these data into the main figure, which would align all figures conceptually.

We have moved autophagy assay to Main Figure 5c, d. We have concluded regarding the result of autophagy assay with a caution based on technical limitation of this assay. The following is actual sentences;

“To assess autophagic activity, we applied tandem fluorescent tagged Atg8a which are widely used in fly to monitor autophagic activity. Systemic knockdown of *dRubicon* significantly increased steady-state autophagosome and autolysosome pools in the brain, suggesting the activation of autophagy (Fig.4c, d). Although in most cases researchers in fly field concluded autophagy activation based on these similar phenotypes, we could not completely rule out the possibility of autophagy inhibition due to, for instance, impaired lysosomal degradation.”

3. All figures should clearly indicate when RNAi was used and all figure legends should clearly describe when the RNAi was initiated (whole-life or adult-only). To this point, the Methods section is missing a description of their double RNAi experiments.

In all experiments, we have started RNAi from egg onward (whole-life) except for TOR knockdown lifespan. Thus, we put this information in each figure legend. We have also described stage of animals in the legends. Method section of double RNAi experiments was also updated.

4. All figures should clearly state when mRNA levels are measured versus protein levels. The nomenclature of CeRubicon is confusing, as it does not adhere to *C. elegans* nomenclature standards. To this end, the authors are encouraged to get a name formally curated in the *C. elegans* field (see Wormbase for instructions).

Thank you so much for the comments. For the nomenclature of *C. elegans* rubicon homolog, we named it *rub-1* based on wormbase instructions.

5. In Figure 5g, the Figure should indicate that the green punctae are alpha-syn.

Now we have inserted this word to indicate green punctae are alpha-syn in Figure 5 legend. Thank you.

6. Line 79-80: the authors mention a recent paper from the Levine lab in the discussion, it should be mentioned in the introduction, along with other studies extending lifespan by autophagy-gene

modulation discussed there.

We have inserted a paper from Levine lab in the introduction as well together with other lifespan studies by activation of autophagy.

7. Line 184: The authors are not testing that increased autophagy in neurons is sufficient to extend lifespan. This needs to be rephrased to say that autophagy gene *atg-18* is required for the lifespan extending effects of neuronal Rubicon reduction.

That is right. We have rephrased to say that longevity is just dependent of *atg-18*. And we have changed the concluding sentence to modest expression, “These results imply that the lifespan extension by *rub-1* knockdown could be due to activation of autophagy.”.

8. Line 299: The authors should delete “ while the knockdown of *bec-1*... “ until the end of the paragraph.

We have removed the whole sentence.

9. Line 306. The authors do not show an “incremental” increase. This should be revised.

We have removed “incremental” from the sentence. Thank you.

Suggestions:

1. Since the authors are not characterizing the CeRubicon overexpressor strains in any other assays besides lifespan, Figure 1 d,h,i could be moved to the supplement.

Thank you for the suggestion. We have now moved the overexpression lifespan to Supplementary Figure 4a,b.

2. Figure 3 could be made into a neuronal-specific figure in which Figure 3b,c,d, are moved into the supplement (noting concerns about rigor/repeats listed above), and Figure S7A is moved into Figure 3.

As you suggested, we have moved previous Figure 3b-d to Supplementary Figure 7 a-c, while we have inserted two neuron specific knockdown lifespan data in Figure 3a and b. Thank you for the constructive suggestion!

3. [redacted]

4. A strain list indicating both previously published and newly generated strains would be helpful.

We have generated a strain list with these descriptions as Supplementary Table 2. Thank you.